# The Psychological Effects of Relational Job Characteristics Scale: An Adaptation Study for Brazilian K-12 Teachers

Natan Klein [1], Carlos Costa [1,*], João P. Marôco [2,3], Cicero Roberto Pereira [4] and Maria José Chambel [5]

1   Graduate Program in Psychology, School of Health, Atitus Educaçãon, Passo Fundo, RS 99070-22, Brazil; natanklein2104@gmail.com
2   William James Center for Research, ISPA—Instituto Universitário, 1149-041 Lisbon, Portugal; jpmaroco@ispa.pt
3   FLU Pedagogy, Nord University, 8026 Bodø, Norway
4   Institute of Social Sciences, University of Lisbon, 1600-189 Lisboa, Portugal; crp@labesp.org
5   CicPsi, Faculdade de Psicologia, Universidade de Lisboa, 1649-013 Lisboa, Portugal; mjchambel@psicologia.ulisboa.pt
*   Correspondence: carlos.costa1@gmail.com

**Abstract:** Relationally enriched jobs can foster psychological states, which, with respect to the beneficiaries of one's work, drive one's motivation. These states can be measured with the psychological effects of relational job characteristics scale, an instrument for which the validity is, at present, only supported by preliminary evidence. Accordingly, the present study's objective was to test a set of psychometric properties reflecting the validity and reliability of the interpretations proposed for this scale. Findings drawn from a sample of 2011 K-12 Brazilian teachers suggested that a tridimensional structure with some residual covariances afforded the best fit. Given the existence of high inter-factor correlations, a second-order factor was proposed as a complementary, if not necessary, feature. Internal consistency omega and alpha coefficients attested to the scores' reliability, and the factor structure achieved indicates invariance across public state, public municipal, and private Brazilian schools. Different relationships with prosocial motivation and work engagement were identified, suggesting validity of the scale based on relationships with other variables.

**Keywords:** psychological effects of relational job characteristics; validity; Brazilian teachers; relational job characteristics

## 1. Introduction

Originally focusing on the characteristics of tasks that affect workers' attitudes and behaviors [1,2], job design theory and research has evolved to address the challenges posed by the emerging trend of knowledge and service economics [3]. Such changes in the workplace draw attention to the relationship workers have with people outside their organization, more specifically, those who are subject to the impacts of their work, i.e., clients, costumers, or beneficiaries [4]. In this manner, modelling and measuring "relational" job characteristics has become an imperative to the advancement of job design research, theory, and practice.

As in the traditional job characteristics model [1,2], jobs have a relational structure that nurtures specific psychological states and fosters attitudinal and behavioral outcomes. Grant's job impact framework provides a theoretical outline of the propositions that support this structure [5]. The framework was built upon quasi-experimental and psychometric evidence found in diverse studies [3–11].

Although a theoretical model and a body of empirical research already existed, a comprehensive, data-driven systematic study aimed at developing a standardized instrument to measure the desired effects of relational workplace characteristics was still

pending. Measures for the intended constructs were built by gathering—in a rather unsystematic manner—different items employed in previous studies. Santos et al. identified this gap and filled it with the first psychometric study on a group of three psychological states fostered by the relational structure of a job, as adapted to Portuguese and Brazilian nurses [12]. Such a measure was termed the psychological effects of relational job characteristics (PERJCs) scale.

Given the relevance of the relational job characteristics topic in job design research, the scarcity of psychometric studies with the PERJCs, and the urgent need to provide sufficient evidence for the validity of constructs measured in psychological research, the main objective of this study is to conduct a comprehensive psychometric evaluation of the PERJCs scale in a sample of Brazilian K-12 teachers. To achieve this, both a unidimensional structure and a three-factor first-order structure are tested, allowing an in-depth investigation of the underlying structure of the scale. Rigorous assessments of convergent, discriminant, and internal consistency reliability estimates are conducted to improve our understanding of the internal structure of the scale. In addition, measurement invariance is examined for different types of schools (public municipal, public, and private) to ensure the applicability of the scale in different educational settings. To establish the validity of the scale, its relationships with other variables, such as prosocial motivation and work engagement, are carefully examined and analyzed. The purpose of this study is to contribute to the advancement of job design, work engagement, and teachers' work research through a comprehensive psychometric evaluation of the PERJCs scale and by establishing its validity and reliability in the context of Brazilian K-12 teachers.

### 1.1. Grant's Job Impact Framework

The primary focus of job design is about defining the fundamental characteristics inherent in the structure of work and understanding their relationship to desired outcomes, such as motivation, wellbeing, and performance [13]. This approach allows us to explore how contextual factors influence employees and how certain psychological states and behaviors manifest under different work conditions. This perspective has been particularly effective in explaining prosocial behavior, which involves making a positive impact on the world, and the motivation underlying this behavior, known as prosocial motivation [6].

To illustrate this point, several field experiments [7,8] used fundraising callers who had the opportunity to interact with beneficiaries of their work. The callers' job was to continually call alumni of a college and ask them to donate to scholarships. This job was characterized by poor working conditions, which was reflected in the high turnover rate among callers. Although callers were unaware of the importance of their work in supporting students through scholarships, connecting with beneficiaries and allowing them to see the impact of their work had a profound impact on their motivation and performance [7,8]. This demonstration underscores the powerful impact of job design on employees when they have a clear understanding of the meaningful difference their work makes.

Given the potential impact of social job characteristics on workers' motivation and propensity to engage in prosocial behavior, it is critical to describe these structural aspects and how they differ across job types. For instance, the work of nurses and teachers has more profound and lasting effects on beneficiaries than the work of industrial managers or telephone operators. Nurses and teachers also have more frequent, longer, physically closer, and more meaningful interactions with patients or students, unlike many other occupations that may have limited contact with their clients. Grant introduced two constructs, work impact on beneficiaries and contact with beneficiaries, to capture variability in opportunities to engage with and influence the lives of others [7]. The former varies in terms of extent, frequency, and scope, while the latter includes frequency, breadth, and depth. Empirical research has shown that these dimensions vary widely in job design [6].

Similar to other job characteristics, the extent to which a work position provides opportunities to influence and interact with beneficiaries influences certain psychological

states, namely other-focused psychological states. These states include perceived social influence (PSI), which reflects the perception that one's work has a positive impact on others, and affective commitment to clients (ACC), which signifies an emotional attachment and devotion to those who benefit from one's work [7]. These constructs have also been shown to be discriminable in psychometric evaluations [6].

Although not originally included in the job impact framework [5], perceived social value (PSW) has been studied as a psychological state that mediates the relationship between work opportunities that benefit others and self-motivation and performance [7]. PSW refers to employees' perceptions that their actions at work are valued by beneficiaries. Recognizing the positive impact of one's work is different from perceiving that this impact is truly valued. Consistent with this perspective, a sense of social value meets a basic psychological need [14] and has significant motivational potential.

These three psychological states, collectively referred to as the psychological effects of relational job characteristics (PERJCs), are thought to promote a specific motivational state known as prosocial motivation [15]. Prosocial motivation involves a desire to benefit others and a willingness to invest time and effort to achieve this [16]. Prosocial motivation has been linked to important personal and organizational outcomes, including wellbeing, prosocial behavior, job performance, and career success, as originally theorized [9].

## 1.2. Studies with the Psychological Effect of Relational Job Characteristics

Several studies have tested the relationships existing between the PERJCs and prosocial motivational, work engagement, burnout, performance, and other similar constructs [17–22]. Although this research followed the theoretical perspective of Adam Grant, as described above, the constructs were measured with different groupings of items. Moreover, few studies [12,20] have evaluated all three PERJCs. In line with Grant's model [10], researchers have commonly only evaluated PSI and ACC. Nonetheless, these studies expanded the nomological network of the PERJCs, demonstrating relationships that extend beyond the initial scope. In Table 1, eight important studies are succinctly described, with a special focus on the relationships with work engagement and prosocial motivation, since these variables are used to test the validity of evidence based on the relationships with other variables of the PERJCs scale.

**Table 1.** Studies with the psychological effects of relational job characteristics scale and associations with prosocial motivation or engagement.

| | Study | Sample | Item's Grouping | *M, SD* | Reliability | β |
|---|---|---|---|---|---|---|
| 1 | Grant (2008a) | 201 public service (lifeguards and police officers) and telephone solicitation employees | PSI: 3 items [7,8]. ACC: 3 items [8] 7-point Likert-type scale Beneficiaries were represented according to each job | PSI: *M* = 5.03, *SD* = 1.06 ACC: *M* = 4.27, *SD* = 1.48 | PSI$_\alpha$ = 0.86 ACC$_\alpha$ = 0.90 | PSI → PSM: 0.22 ACC → PSM: 0.51 |
| 2 | Freeney and Fellenz (2013) | 182 Irish Midwives | PSI: 4 items [7] 7-point Likert-type scale Beneficiaries were represented as patients | PSI: *M* = 6.20, *SD* = NA | PSI$_\alpha$ = 0.82 | PSI → ENG: 0.32 *** |
| 3 | Castanheira (2016) | 370 customer service employees (bankers, retailers, and callers) | PSI: 6 items [7,10]. PSW: 3 items [7]. ACC: 3 items [6] 7-point Likert-type scale Beneficiaries were represented as customers Items were translated into Portuguese (Portugal) | PSI: *M* = 4.36, *SD* = 1.27 PSW: *M* = 3.89, *SD* = 1.38 ACC: *M* = 5.27, *SD* = 1.21 | PSI$_\alpha$ = 0.89 PSW$_\alpha$ = 0.83 ACC$_\alpha$ = 0.89 PSI$_{CR}$ = 0.90 PSW$_{CR}$ = 0.83 ACC$_{CR}$ = 0.90 | PSI → ENG: 0.32 *** PSW → ENG: 0.12 n.s. ACC → ENG: 0.13 * |

**Table 1.** *Cont.*

| | Study | Sample | Item's Grouping | *M, SD* | Reliability | β |
|---|---|---|---|---|---|---|
| 4 | Santos et al. (2016a) | 335 hospital nurses | PSI: 6 items [7,10]. PSW: 3 items [7]. ACC: 2 items [6], one was excluded for redundancy 7-point Likert-type scale Beneficiaries were represented as "others" | PSI: $M$ = 5.85, $SD$ = 0.82 PSW: $M$ = 5.14, $SD$ = 1.17 ACC: $M$ = 5.46, $SD$ = 1.11 | $PSI_\alpha$ = 0.93 $PSW_\alpha$ = 0.91 $ACC_\alpha$ = 0.85 | PSI → ENG: 0.17 ** PSW → ENG: 0.16 * ACC → ENG: 0.22 ** |
| 5 | Castanheira et al. (2016) | 322 Officers and Sergeants | Only PSI and PSW, same as study 3 | PSI: $M$ = 5.27, $SD$ = 1.09 PSW: $M$ = 4.62, $SD$ = 1.38 | $PSI_\alpha$ = 0.95 $PSW_\alpha$ = 0.90 | PSI → ENG: 0.37 *** PSW → ENG: 0.17 * PSI → PSM: 0.62 *** PSW → PSM: −0.10 n.s. |
| | | 1045 soldiers | | PSI: $M$ = 4.52, $SD$ = 1.28 PSW: $M$ = 4.36, $SD$ = 1.42 | $PSI_\alpha$ = 0.93 $PSW_\alpha$ = 0.86 | PSI → ENG: 0.27 *** PSW → ENG: 0.23 *** PSI → PSM: 0.34 *** PSW → PSM: 0.09 * |
| 6 | Santos et al. (2017b) | 335 Portuguese hospital nurses | PSI, PSW, and ACC, same as study 3 | PSI: $M$ = 5.85, $SD$ = 0.82 PSW: $M$ = 5.14, $SD$ = 1.17 ACC: $M$ = 5.46, $SD$ = 1.11 | $PSI_\alpha$ = 0.93 $PSW_\alpha$ = 0.91 $ACC_\alpha$ = 0.85 | PSI → ENG: 0.15 * PSW → ENG: 0.24 ** ACC → ENG: 0.15 * |
| | | 285 Brazilian hospital nurses | | PSI: $M$ = 6.09, $SD$ = 0.78 PSW: $M$ = 5.75, $SD$ = 1.13 ACC: $M$ = 5.58, $SD$ = 1.06 | $PSI_\alpha$ = 0.92 $PSW_\alpha$ = 0.88 $ACC_\alpha$ = 0.73 | PSI → ENG: NA PSW → ENG: 0.36 ** ACC → ENG: NA |
| 7 | Santos et al. (2017a) | 620 hospital nurses (335 Portuguese and 285 Brazilian) | PSI, PSW, and ACC, same as study 3 | PSI: $M$ = 5.95, $SD$ = 0.81 PSW: $M$ = 5.31, $SD$ = 1.11 ACC: $M$ = 5.60, $SD$ = 1.14 | $PSI_\alpha$ = 0.93 $PSW_\alpha$ = 0.89 $ACC_\alpha$ = 0.83 | PSI ↔ ENG: 0.36 ** PSW ↔ ENG: 0.39 ** ACC ↔ ENG: 0.32 ** |
| 8 | Santos et al. (2020) | 409 Portuguese hospital nurses | PSI, PSW, and ACC, same as study 3 | PSI: $M$ = 6.02, $SD$ = 0.88 PSW: $M$ = 5.07, $SD$ = 1.26 ACC: $M$ = 5.65, $SD$ = 1.14 | $PSI_\alpha$ = 0.94 $PSW_\alpha$ = 0.89 $ACC_\alpha$ = 0.86 | PSI → ENG: 0.25 *** PSW → ENG: 0.27 *** ACC → ENG: 0.05 n.s. |

Note. NA = not available; PSI = perceived social impact; ACC = affective commitment to clients; PSW = perceived social worth; ENG = work engagement; PSM = prosocial motivation. n.s. nonsignificant; * $p \le 0.05$, ** $p \le 0.01$, *** $p \le 0.001$.

### 1.3. The Present Study

Santos et al. [12] evaluated the psychometric properties of the PERJCs scale with a sample of Brazilian and Portuguese nurses and demonstrated a satisfactory adjustment to the three-dimensional model with invariance across Portuguese and Brazilian samples. In the present study, we aimed to examine the dimensionality of the PERJCs scale in a sample of Brazilian K-12 teachers. We hypothesized that the three-factor structure would

provide a better fit compared to a unidimensional structure, given the subtle but significant differences between the constructs being assessed. We expected high factor loadings, consistent with previous studies, indicating strong relationships between the items and their respective factors. In addition, we expected that the scale would have satisfactory internal consistency and discriminability among the three dimensions, a property that has not been extensively examined in previous studies.

Because the teaching profession in Brazil, similar to hospital nursing, has a high relational component, we expected that Brazilian teachers would have high scores on the PERJCs scale. In addition, we hypothesized that the factor structure would be invariant across different school types, including public municipal, public, and private schools. Furthermore, we expected positive associations between the PERJCs scale and both prosocial motivation and work engagement. However, we predicted stronger coefficients for the association with prosocial motivation, consistent with the hypotheses put forth by Grant [9].

## 2. Materials and Methods

### 2.1. Participants

A convenience sample of 2202 Brazilian K-12 teachers, working in 16 municipalities of Rio Grande do Sul state, were asked to complete a questionnaire. Of these, only 2011 questionnaires were considered in this study as the remainder were inconsistent or incomplete. Respondents worked at public municipal (82.0%, $n = 1650$), public state (7.9%, $n = 159$), and private schools (10.0%, $n = 202$). Overall, teachers taught in kindergarten (33.0%, $n = 879$), elementary school (56.6%, $n = 1507$), and high school (10.4%, $n = 275$). Participants were mostly women (92.7%, $n = 1853$), with an average age of 40.24 ($SD = 9.72$, $Min = 20$, $Q_1 = 33$, $Q_2 = 39$, $Q_3 = 48$, $Max = 84$) and 14.92 ($SD = 9.27$) years of experience in education. These teachers' highest educational level was undergraduate school (84.5%, $n = 1681$), graduate school ("specialization", neither master's nor doctorate; 5.9%, $n = 118$), master's degree (7.2%, $n = 143$), or doctoral degree (2.4%, $n = 48$). These teachers worked an average of 28.66 ($SD = 11.85$) hours per week.

### 2.2. Procedures

Printed questionnaires were given to participating teachers during their working hours or at teacher meetings at the schools. Prior to data collection, arrangements were made with municipal education departments (for municipal public schools) and regional education coordinators (for state public schools) to ensure adequate coordination. Meetings were held with school principals to explain the objectives of the study and to obtain permission to conduct the study in their schools. To ensure the confidentiality of the data and protect the anonymity of the participants, the information collected was kept confidential and stored securely. In accordance with Brazilian laws governing ethical standards for research involving human subjects, an informed consent form was given to each participant before completing the questionnaires. This consent form explained the purpose of the study, the voluntary nature of participation, and the protection of personal data. Completion of the questionnaires took approximately 20 min per participant. The research procedures and protocols were reviewed and approved by the local research and ethics committee to ensure compliance with ethical guidelines.

### 2.3. Instruments

2.3.1. Psychological Effects of Relational Job Characteristics (PERJCs) Scale

The PERJCs scale was composed of three subscales constructed for and applied in different studies [6,7,11]. The items of these subscales were further grouped and adapted to the Portuguese language both for Brazil and Portugal by Santos et al. [12]. The perceived social impact factor was composed of six items, the perceived social worth factor was based on three items, and the affective commitment to the client dimension was measured with two items. English and Portuguese items, as well as the response options (7-point Likert-type), are presented in Supplementary Table S1.

### 2.3.2. Prosocial Motivation Scale (PSMS)

The PSMS was composed of items used to measure prosocial motivation in different studies [6,11,19] The version adapted for the Brazilian context consisted in a revision of a Portuguese version used in previous studies [17,18]. The instrument measures a unidimensional prosocial motivation construct with 12 items. English and Portuguese items, as well as the response options (7-point Likert-type), are presented in Supplementary Table S2.

### 2.3.3. Utrecht Work Engagement Scale—Short Version (UWES-9)

The UWES-9 was devised by Schaufeli et al. [23] and their version, adapted for a Brazilian context, had been used in previous transcultural studies undertaken in both Portugal and Brazil [12,20]. This instrument measures three dimensions of work engagement, each with three items: vigor, dedication, and absorption. A previous study reported that a model with an additional second-order work engagement general factor achieved the best fit in Portuguese and Brazilian samples [24]. English and Portuguese items, as well as the response options (7-point Likert-type), are presented in Supplementary Table S3.

### *2.4. Data Analysis*

All statistical analyses were performed using R v.4.22 [25] with the R Studio interface [26]. There were no missing values in the data because listwise deletion was performed before analysis. Descriptive statistics were computed using the skimr package [27].

To test the unidimensional and three-dimensional factor structure of the PERJCs scale, confirmatory factor analysis (CFA) was performed using the Lavaan package [28]. The robust maximum likelihood estimator (MLR) was used because the observed indicators were measured on 6 to 7 Likert scales [29]. Model fit was considered good when the ratio of chi-squared to degrees of freedom ($\chi^2/df$) was <5, the confirmatory fit index (CFI) and the Tucker–Lewis index (TLI) were $\geq$0.95, and the root mean square error of approximation (RMSEA) was <0.08 [30–33].

Estimates of internal consistency reliability for the measures were calculated using alpha ($\alpha$) and omega ($\omega$) coefficients [34,35]. Evidence for convergent and discriminant validity of the PERJCs scale was assessed using the average variance extracted (AVE). Adequate convergence was assumed when AVE $\geq$ 0.5 [36], and when AVE $\geq \rho^2_{xy}$ (the squared correlation between two factors), adequate discrimination between constructs was established.

To evaluate the invariance of the PERJCs structure across groups of teachers from different types of schools (public municipal, public, and private schools), the Lavaan package was used. Four different models were compared: configural, fixed factor loadings, fixed intercepts, and fixed means. Fit indices were examined for each model, with particular attention to the difference in CFI ($\Delta$CFI) between models. A decrease in the CFI > 0.01 ($\Delta$CFI) indicates a lack of invariance [37].

Validity evidence based on relationships with other variables (nomological validity) was obtained through a structural model in which a second-order work engagement factor and a first-order prosocial motivation factor were predicted by the three PERJC factors. The estimation and fit assessment of this model followed the same procedure as the CFA described above.

## 3. Results

The psychological effects of relational job characteristics (PERJCs) scale was assessed in terms of its reliability and the validity evidence of its scores. As a basis for such evidence, we first describe the item's characteristics. Then, we report validity evidence based on the internal structure, evaluating the fit of the hypothesized models, convergent and discriminant evidence, and the factor structure invariance across different kinds of schools. Next, we outline validity evidence based on the relationship with other variables, namely, prosocial motivation and work engagement.

### 3.1. Item's Characteristics

Descriptive measures, skewness, and kurtosis values, as well as a histogram are presented in Table 2 for each item. Absolute Skewness values were <3 and kurtosis values were <7, which suggests no strong deviation from a normal distribution [38]. Nonetheless, means and the histograms indicate a tendency toward ceiling effects. This could suggest that the evaluated sample displays a high level on the latent trait and/or items are too "easy" for the assessed sample, i.e., the trait level item's capture did not match with the sample's trait level. Furthermore, there were no empty categories in the item's Likert-type scale.

**Table 2.** The PERJCs items' characteristics.

| Item | *M* | *SD* | Skewness | Kurtosis | Histogram |
|---|---|---|---|---|---|
| PSI_1 | 6.18 | 0.83 | −1.41 | 4.18 | |
| PSI_2 | 6.09 | 0.88 | −1.24 | 2.99 | |
| PSI_3 | 6.06 | 0.92 | −1.19 | 2.47 | |
| PSI_4 | 5.85 | 0.94 | −0.88 | 1.51 | |
| PSI_5 | 5.81 | 0.97 | −1.06 | 2.26 | |
| PSI_6 | 5.70 | 1.05 | −0.96 | 1.60 | |
| ACC_1 | 6.05 | 1.27 | −2.20 | 5.43 | |
| ACC_2 | 5.93 | 1.14 | −1.60 | 3.30 | |
| PSW_1 | 5.60 | 0.98 | −1.08 | 2.31 | |
| PSW_2 | 5.41 | 1.11 | −1.16 | 2.05 | |
| PSW_3 | 5.63 | 1.09 | −1.47 | 3.10 | |

Note. PSI = perceived social impact; ACC = affective commitment to clients; PSW = perceived social worth. For all items, minimum values were 1, maximum values were 7, and median values were 6.

### 3.2. Validity Evidence Based on the Internal Structure

3.2.1. Factor Structure and Internal Consistency of the PERJCs

To determine the most appropriate factor structure for our data, we tested several models in order of increasing complexity. Table 3 shows the fit indices we obtained for the following models: a unidimensional model, a tridimensional model, and a tridimensional model with residual covariances included.

**Table 3.** Fit indices for the unidimensional, tridimensional, and tridimensional with residual covariances models.

| Factor Models | $\chi^2_{(df)}$ | $\chi^2/df$ | $\Delta\chi^2$ | RMSEA 90% CI [LB, UB] | TLI | CFI |
|---|---|---|---|---|---|---|
| Unidimensional | 2877.599 * $_{(44)}$ | 65.40 | - | 0.179 [0.173, 0.185] | 0.684 | 0.747 |
| Tridimensional | 627.911 * $_{(41)}$ | 15.30 | 2249.689 * | 0.084 [0.079, 0.090] | 0.930 | 0.948 |
| Tridimensional with residual covariances | 189.220 * $_{(44)}$ | 4.30 | 438.691 * | 0.044 [0.038, 0.051] | 0.980 | 0.987 |

Note. The model fit was considered good when the $\chi^2/df$ ratio was <5, the CFI (confirmatory fit index) and TLI (Tucker–Lewis index) was ≥0.95, and the RMSEA (root mean square error of approximation) was <0.08. LB = lower bound; UB = upper bound. * $p \leq 0.001$.

Despite factor loadings above 0.30, the unidimensional model did not fit the data well. The theorized tridimensional model showed a significantly better fit, although it did not reach the level of fit expected for its definitive acceptance. To improve the fit, modification indices were investigated. These indices suggested the inclusion of residual covariances between certain indicators of the PSI factor. This could be due to the nature of the items, as they capture slightly different aspects of the PSI construct but share the dominant theme of perceiving a positive impact on others. As a result, residual covariances were included

in the model, resulting in the best fit. The factor loadings for the tridimensional structure with the residual covariances implemented are shown in Figure 1.

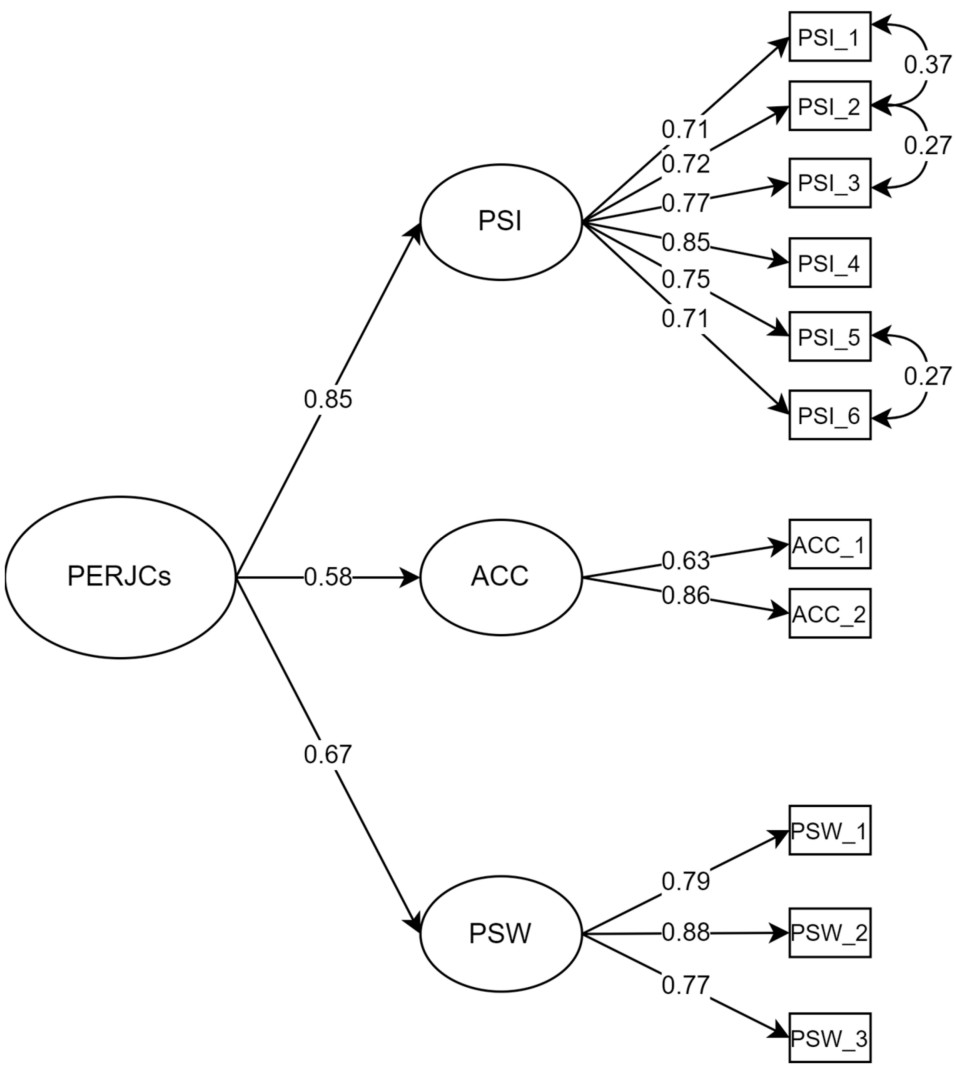

**Figure 1.** The psychological effects of relational job characteristics scale factor structure. PSI = perceived social impact; ACC = affective commitment to clients; PSW = perceived social worth. All factor loadings and covariances were statistically significant at $p \leq 0.001$.

Figure 1 shows that factors' covariances were moderate to high, suggesting the suitability of a second-order model. In fact, the three PERJCs have task significance as their common theme, although with the "social" aspect emphasized over other characteristics. Its addition enables us only to evaluate the second-order factor loadings but not the impact of the structural increment on the structure's fit. This way, the implementation of a second-order factor does not alter (estimate) the fit indices. The second-order loadings were: PERJCs → PSI ($\lambda = 0.85$), PERJCs → ACC ($\lambda = 0.58$), PERJCs → PSW ($\lambda = 0.67$). The magnitude of these loadings highlights that computation of a general PERJCs scale is a viable, although not ideal, alternative.

Moreover, regarding the reliability of the three-factor solution, alpha, $\alpha_{PSI} = 0.926$, $\alpha_{ACC} = 0.836$, $\alpha_{PSW} = 0.884$, as well as omega values, $\omega_{PSI} = 0.904$, $\omega_{ACC} = 0.766$, $\omega_{PSW} = 0.846$, indicated satisfactory internal consistency. In a similar way, $\omega_{PERJCs} = 0.764$, suggested adequate internal consistency reliability for a second-order latent factor.

### 3.2.2. Convergent and Discriminant Validity Evidence

The average variance extracted (AVE) was used to assess the degree to which items converge on their presumed factor. Good convergent validity evidence can be assumed, since AVE $\geq$ 0.5 (Table 4). Furthermore, the results also indicated good discriminant validity evidence, since AVE values were greater than the squared correlations between factors ($\rho^2_{xy}$).

**Table 4.** Average variance extracted and the squared correlations between the factors of the psychological effects of the relational job characteristics scale.

|  | PSI | ACC | PSW |
|---|---|---|---|
| PSI | 0.697 | | |
| ACC | 0.230 | 0.687 | |
| PSW | 0.302 | 0.152 | 0.777 |

Note. PSI = perceived social impact; ACC = affective commitment to the clients; PSW = perceived social worth. Average variance extracted (AVE) values are presented in the diagonal and squared correlations between factors in the corresponding cases.

### 3.2.3. Measurement Invariance across the Different Kinds of Schools

A series of four models were tested in order to evaluate the measurement invariance of the PERJCs across different kinds of schools (public municipal, public state, and private). Fit indices for each model attest its proper fit (Table 5). Moreover, the DCFI test based on the criteria ($\Delta$CFI > 0.01 = invariance rejected) [38] indicates a complete invariance of the factor model across different kinds of schools, i.e., means invariance were achieved.

**Table 5.** Sequential models for testing the psychological effects of relational job characteristics scale's measurement invariance across three different kinds of schools.

|  | $\chi^2$ | df | RMSEA | TLI | CFI | $\Delta$CFI |
|---|---|---|---|---|---|---|
| Configural | 292.767 | 114 | 0.048 | 0.977 | 0.984 | - |
| Fixed factor loadings | 319.636 | 136 | 0.045 | 0.980 | 0.984 | - |
| Fixed intercepts | 351.076 | 152 | 0.044 | 0.981 | 0.982 | −0.002 |
| Fixed means | 362.007 | 158 | 0.044 | 0.981 | 0.982 | 0.000 |

Note. $\Delta$CFI > 0.01 was taken as a criterion to reject invariance.

To provide data for norming and future studies reference, raw mean and standard-deviation scores for the general sample and each kind of schools are presented in Table 6, and decile scores for the general sample is provided in Table 7. As the fixed means model already demonstrated, the scores do not show a statistically significant difference across school type. Moreover, in general, Brazilian teachers exhibited high scores on all the PERJC dimensions.

**Table 6.** Raw mean and standard deviation scores for the whole sample and public municipal, public state, and private schools.

|  | Whole Sample $n$ = 2011 | | Public Municipal $n$ = 1650 | | Public State $n$ = 159 | | Private $n$ = 202 | |
|---|---|---|---|---|---|---|---|---|
|  | *M* | *SD* | *M* | *SD* | *M* | *SD* | *M* | *SD* |
| PSI | 5.95 | 0.76 | 5.95 | 0.75 | 5.88 | 0.79 | 6.05 | 0.77 |
| ACC | 5.99 | 1.06 | 6.01 | 1.03 | 5.99 | 1.17 | 5.90 | 1.19 |
| PSW | 5.55 | 0.93 | 5.53 | 0.94 | 5.56 | 1.04 | 5.68 | 0.73 |
| PERJCs | 5.85 | 0.68 | 5.84 | 0.68 | 5.81 | 0.75 | 5.92 | 0.66 |

Note. PSI = perceived social impact; ACC = affective commitment to clients; PSW = perceived social worth; PERJCs = psychological effects of relational job characteristics. Min = 1; Max = 7.

**Table 7.** Decile scores for the PERJCs scale's three factors and general factor.

| Deciles | PSI | ACC | PSW | PERJCs |
|---------|-----|-----|-----|--------|
| 10 | 5.00 | 4.50 | 4.33 | 5.00 |
| 20 | 5.33 | 5.50 | 5.00 | 5.36 |
| 30 | 5.67 | 6.00 | 5.33 | 5.55 |
| 40 | 5.83 | 6.00 | 5.33 | 5.73 |
| 50 | 6.00 | 6.00 | 5.67 | 5.91 |
| 60 | 6.17 | 6.50 | 6.00 | 6.09 |
| 70 | 6.33 | 7.00 | 6.00 | 6.18 |
| 80 | 6.67 | 7.00 | 6.00 | 6.45 |
| 90 | 7.00 | 7.00 | 6.67 | 6.73 |

Note. PSI = perceived social impact; ACC = affective commitment to the clients; PSW = perceived social worth; PERJCs = psychological effects of relational job characteristics. Min = 1; Max = 7.

3.2.4. Validity Evidence Based on Relationships with Other Variables

The validity evidence based on the relationship with other variables was assessed with a structural model where PSI, ACC, and PSW predicts prosocial motivation and work engagement. The model showed a good fit, $\chi^2_{(448)}$ = 3222.267, $\chi^2/df$ = 7.19, CFI = 0.924, TLI = 0.915, RMSEA = 0.055, 90% CI [0.054, 0.057]. Covariances within the PERJCs model were maintained, and the covariance between the prosocial motivation and work engagement factors was, $\text{cov}_{\text{PSM-ENG}}$ = −0.045, $p$ = 0.166. Standardized coefficients are presented in Table 8.

**Table 8.** Regression coefficients between the psychological effects of relational job characteristics, work engagement, and prosocial motivation.

| Prosocial Motivation | B | SE | $p$ | β |
|----------------------|---|-----|-----|---|
| PSI → PSM | 0.339 | 0.043 | <0.001 | 0.432 |
| ACC → PSM | 0.145 | 0.022 | <0.001 | 0.255 |
| PSW → PSM | −0.015 | 0.018 | 0.401 | −0.026 |
| **Work Engagement** | | | | |
| PSI → ENG | 0.118 | 0.047 | 0.011 | 0.101 |
| ACC → ENG | 0.049 | 0.028 | 0.079 | 0.058 |
| PSW → ENG | 0.319 | 0.039 | <0.001 | 0.362 |

Note. PSI = perceived social impact; ACC = affective commitment to clients; PSW = perceived social worth; PSM = prosocial motivation; ENG = work engagement.

The relationships analyzed in Table 8 suggests that PSI is the main driver of prosocial motivation, endorsing the job impact framework's propositions [5]. Moreover, ACC played an important, although secondary role in explaining prosocial motivation. The PSW, however, showed a nonsignificant association. In a different fashion, nevertheless, PSW was the strongest predictor of work engagement, with PSI playing a minor role in the explanation, and ACC showing a nonsignificant association.

**4. Discussion**

Prosocial motivation and work engagement were positively correlated with the three PERJCs scale dimensions, suggesting that teachers who perceived higher levels of relational job characteristics were more likely to engage in prosocial behavior and be engaged in their work. These findings suggest that the psychological effects of relational job characteristics play an important role in shaping the experiences of Brazilian K-12 teachers. The three dimensions of the PERJCs scale, along with its possible second-order dimension, provide a comprehensive framework for understanding the impact of relational job characteristics on teacher wellbeing and job-related outcomes. The results also highlight the importance of considering the specific context of different types of schools when ex-

amining the relationship between relational job characteristics and teacher motivation and engagement.

As far as we know, this was the second study to examine the internal structure of the PERJCs scale. Similar to the first validation study by Santos et al., our results rejected a unidimensional model and supported a tridimensional structure [12]. The fit indices we obtained were even comparable to those of Santos et al. [12], but we achieved a reduction in error by implementing residual covariances, resulting in lower RMSEA values. In addition, our analysis consistently yielded high factor loadings, similar to previous results.

The internal consistency indices, alpha and omega, confirmed the reliability of scores derived from the three dimensions, with magnitudes similar to those reported in other studies (see Table 1). Moreover, the coefficients correspond to the order of reliability observed in the literature, with PSI being the most reliable dimension, followed by PSW and then ACC in decreasing order. Although the order of reliability has no practical significance, the PSI dimension proved to be the most important facet for the overarching PERJC factor when second-order factor loadings were considered. Thus, if we interpret the second-order dimension as task significance from a social or other-focused perspective, PSI would be the most important component.

Regarding the internal structure of the instrument, future studies could investigate the exclusion of certain PSI items, as not all structural models have sufficient degrees of freedom to estimate three residual covariances, as we did. Nonetheless, the present evidence on the dimensionality of the PERJCs scale is encouraging, although further studies with different samples are needed to assess its replicability.

While some studies point to differences in wellbeing indicators (work-related quality of life and burnout) between Brazilian teachers in public and private schools [39,40], our results suggest invariance in the PERJCs scale across school types, especially in terms of latent means (i.e., nonsignificant mean differences). This could be due to the fact that PERJCs results from professional characteristics are inherent to each teacher's job and do not differ across school types. Moreover, PERJCs dimensions are primarily antecedents of motivational states. If they were primarily antecedents, indicators, or consequences of states of wellbeing/dissatisfaction, we would expect more variability across school types.

Comparing the PERJC mean scores found in our sample to those reported in Table 1, we can surmise that teachers' scores are similar to those of nurses and higher than those of other occupational groups such as soldiers and customer service employees. However, these observations are preliminary and require further testing in subsequent studies.

Regarding the relationships between the PERJCs scale and prosocial motivation, our results are consistent with the theoretical literature, which considers the PSI dimension to be the strongest predictor [9]. However, it should be noted that Grant [6] found the opposite in his study and Castanheira et al. [18] found different coefficients for the influence of PSI on prosocial motivation in different samples. Unfortunately, due to the lack of other studies that have examined the relationships between the PERJCs scale and prosocial motivation, we lack interpretive anchors for our results. We hope that our results, along with future studies, will contribute to a clearer understanding of these relationships.

Regarding the associations between the PERJCs scale and work engagement, PSW emerged as the strongest predictor, followed by PSI. However, there is no consensus on the magnitude of the associations between the PERJCs scale and work engagement, as shown by the studies listed in Table 1. In our sample, PSW had a similar magnitude to hospital nurse samples [21], but the coefficients of PSI were generally lower compared with other studies. While many studies have found significant associations between ACC and work engagement, our analysis did not replicate this pattern in our sample. For the Brazilian teachers we studied—and perhaps for a larger group of Brazilian K-12 teachers—the social value of their work appeared to be more important in promoting professional engagement than their perceptions of the impact of their work on students.

However, a limitation of this study was that the invariance in the factor structure between genders could not be assessed due to the small sample size of male teachers

(only 158 participants). In order to obtain reliable estimates, a larger sample would be needed to examine gender differences. In addition, the antecedents of the PERJCs scale, such as influence and contact, were not measured in this study, which could provide further insight. In future studies, it would be important to investigate whether objective opportunities and their psychological effects can be distinguished, as suggested by Grant [6]. In addition, future research could investigate the underlying factors that contribute to the different patterns of relationships between the PERJCs scale and variables such as prosocial motivation, work engagement, and other motivation-related constructs.

## 5. Conclusions

The present study examined several psychometric properties to assess the validity and reliability of the psychological effects of relational job characteristics (PERJCs) scale in a sample of Brazilian K-12 teachers. Overall, the results provided evidence for the validity of the PERJCs scale in measuring the dimensions of perceived social impact (PSI), affective commitment to clients (ACC), and perceived social worth (PSW), as well as a second-order factor of task significance. Results indicated that the scale provided reliable scores that could be compared with different types of Brazilian schools, including private, state, and municipal schools.

**Supplementary Materials:** The following supporting information can be downloaded at: https://www.mdpi.com/article/10.3390/merits3040040/s1, Table S1: Psychological effects of relational job characteristics scale items in English and Portuguese; Table S2: Prosocial motivation scale items in English and Portuguese; Table S3: Utrecht work engagement scale—nine items in English and Portuguese.

**Author Contributions:** Conceptualization, C.C., J.P.M., M.J.C. and C.R.P.; methodology, J.P.M.; formal analysis, J.P.M. and N.K.; investigation, C.C.; writing—original draft preparation, N.K.; writing—review and editing, C.C., J.P.M., C.R.P. and M.J.C. All authors have read and agreed to the published version of the manuscript.

**Funding:** This research was funded by National Council for Scientific and Technological Development (CNPq), Brazil, grant number 434048/2018-6.

**Institutional Review Board Statement:** This study was conducted in accordance with the Declaration of Helsinki and approved by the Ethics Committee of Atitus Education (62574216.0.0000.5319, 1st December 2016).

**Informed Consent Statement:** Informed consent was obtained from all subjects involved in this study.

**Data Availability Statement:** The data presented in this study are available on request from the corresponding author.

**Acknowledgments:** Carlos Costa thanks the IMED Foundation for personal financial support.

**Conflicts of Interest:** The authors declare no conflict of interest. The funders had no role in the design of the study; in the collection, analyses, or interpretation of data; in the writing of the manuscript; or in the decision to publish the results.

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
