# Peer review of "The Psychological Effects of Relational Job Characteristics Scale: An Adaptation Study for Brazilian K-12 Teachers"

_merits, doi:10.3390/merits3040040_

Round 1
Reviewer 1 Report
Comments and Suggestions for Authors
I would like to thank you very much for the opportunity to review the article.
Reading such a complete and good article based on exciting research was a great pleasure.
The manuscript has a lot of solid points and needs no significant changes.
The only change I suggest to the authors is to divide the Participants and Procedures subsection into smaller parts. It is because it contains information on Ethical considerations, Characteristics of the subjects, Methods, and techniques for implementing the study.
On this occasion, it is unclear to me - due to the condensation of information in this subsection - what was the sampling of respondents.
In the Conclusions chapter, I propose to exclude the Limitations of the Study section and place it as a subsection in the Discussion section.
Author Response
Comments from Reviewer 1
Comment 1: The only change I suggest to the authors is to divide the Participants and Procedures subsection into smaller parts. It is because it contains information on Ethical considerations, Characteristics of the subjects, Methods, and techniques for implementing the study.
Response: Thank you for pointing this out. We agree with this comment. Therefore, we have divided the participants and procedures section into two separate sections
Comment 2: On this occasion, it is unclear to me - due to the condensation of information in this subsection - what was the sampling of respondents.
Response: Thank you for pointing this out. We understand that you were asking about the sample size and sampling method used in our study. Both of these are described in the Methods section, specifically in the first line of the section on participants, where we state that a random sample was used.
Comment 2: In the Conclusions chapter, I propose to exclude the Limitations of the Study section and place it as a subsection in the Discussion section.
Response: We agree with this and have incorporated your suggestion throughout the manuscript. We have moved the limitations paragraph to the end of the discussion section to better convey the overall significance of our findings.
Reviewer 2 Report
Comments and Suggestions for Authors
Thank you for allowing me to review the paper "The Psychological Effects of Relational Job Characteristics Scale: An Adaptation Study for Brazilian K-12 Teachers."
The article is interesting and well written. Following are some comments/suggestions:
- It is not clear to me why the authors include studies from the literature in the table. Perhaps the table could be moved as an appendix. In the literature section, the authors should make an effort to describe the studies by highlighting their strengths and weaknesses with respect to their study aims.
- The criteria for recruiting schools is unclear
- it is unclear also the criteria used for skewness and kurtosis
- there are too many correlations between items in the PSI dimension. Please justify (perhaps even considering the content of the items in the dimension) and a comparison with previous validations of this measure in other contexts (or the original one).
Author Response
Comment 1: It is not clear to me why the authors include studies from the literature in the table. Perhaps the table could be moved as an appendix. In the literature section, the authors should make an effort to describe the studies by highlighting their strengths and weaknesses with respect to their study aims.
Response: You have raised an important point here. However, we included the table in the introduction section for two reasons:
(1) To place our results in the context of the few previous studies that have used this instrument, and
2) The practice of presenting in the introductory section a table summarizing the psychometric properties of the instrument under study has proved useful to us and to other researchers. In other words, a useful tool for other researchers, as exemplified by Sinval et al. (2018).
Sinval, J., Queirós, C., Pasian, S., & Marôco, J. (2019). Transcultural Adaptation of the Oldenburg Burnout Inventory (OLBI) for Brazil and Portugal. Frontiers in Psychology, 10. https://doi.org/10.3389/fpsyg.2019.00338
Comment 2: The criteria for recruiting schools is unclear
Response: Thank you for pointing this out. As indicated in the first line of the section on participants, schools were recruited on the basis of random sampling.
Comment 3: it is unclear also the criteria used for skewness and kurtosis
Response: We use common guidelines in the field of Structural Equation Modeling - SEM (see, e.g., Kline, 2003 or Finney & DiStefano). According to these guidelines, Maximum Likelihood methods can be used in SEM when the absolute values of Skewness and Kurtosis are less than 3 and 7-8, respectively. The bias caused by the non-normality of the item distributions is not large for skewness and kurtosis are within this range.
Comment 4: there are too many correlations between items in the PSI dimension. Please justify (perhaps even considering the content of the items in the dimension) and a comparison with previous validations of this measure in other contexts (or the original one).
Response: Thank you for this comment and the suggestion to understand the residual correlation between the items of the PSI factor. Specifically, we found a low residual correlation between three pairs of items: PSI1-PSI2; PSI2-PSI3; and PSI5-PSI6. Because these items loaded strongly on the common latent factor, it is more likely that their residual association is due to similarities in item wording, as SEM is very sensitive to this aspect. Indeed, PSI1 and PSI2 began with "I am very"; PSI2 and PSI3 have in common that "my work" is passively voiced; and PSI5-PSI6 have in common the ative voice of the subject of the action ("I have"; "My work has"). It is important to note that the low correlation and the fact of error measurement were controlled for in the estimation of the latent variable models. The estimated parameters of the models testing our hypothesis were not affected by these elements.”
Round 2
Reviewer 2 Report
Comments and Suggestions for Authors
The article is accepted in its current form
Author Response
Comment 1: In Data Analysis section there are some typos (e.g. (RMSEA) < was 0.008 instead “was < 0.08”; degrees of freedom (χ²/df) < was 5 instead “was < 5) and so on
Response: We have corrected these typos.
Comment 2: Regards the “Validity Evidence Based” I suggest to specify what kind of validity Authors refer. Could be Nomological?
Response: Thank you for this hint. Yes, it is (nomological validity). We have made the changes.
Comment 3: Futhermore, please check the References section carefully. I noticed many references that do not follow the "MDPI Reference List and Citation Style Guide". You can check them at https://www.mdpi.com/authors/references [List of changes]
Response: We have carefully revised the "References" section and made some minor adjustments.
We have also addressed the issue of self-citations, as you requested. We have reduced the number of self-citations in the first revision previously submitted. We believe that the remaining self-citations are necessary to support our arguments.
We understand that you may wish to reduce the number of self-citations further. If this is the case, we are willing to work with you to find other works that can be cited in place of the authors we have cited.
We would also like to mention that we have not received any further feedback from the first reviewer. However, we have received a statement from the second reviewer in which he mentions that he has approved the paper with the changes we made in the first round of revisions. We believe that the changes we made address all the concerns raised by the reviewers. We are confident that our manuscript is now ready for publication in your journal.